# Genome-wide identification of mitogen-activated protein kinase (MAPK) cascade and expression profiling of *CmMAPKs* in melon (*Cucumis melo* L.)

Xiaolan Zhang[1,2,3☯], Yuepeng Li[1,2,3☯], Qiaojuan Xing[1,2,3], Lingqi Yue[1,2,3], Hongyan Qi [1,2,3]*

1 College of Horticulture, Shenyang Agricultural University, Shenyang, Liaoning, China, 2 Key Laboratory of Protected Horticulture of Education Ministry and Liaoning province, Shenyang, Liaoning, China, 3 Northern National & Local Joint Engineering Research Center of Horticultural Facilities Design and Application Technology, Shenyang, Liaoning, China

☯ These authors contributed equally to this work.
* qihongyan@syau.edu.cn, hyqiaaa@126.com

**Data Availability Statement:** All relevant data are within the paper and its Supporting Information files.

## Abstract

Mitogen-activated protein kinase (MAPK) is a form of serine/threonine protein kinase that activated by extracellular stimulation acting through the MAPK cascade (MAPKKK-MAPKK-MAPK). The MAPK cascade gene family, an important family of protein kinases, plays a vital role in responding to various stresses and hormone signal transduction processes in plants. In this study, we identified 14 *CmMAPKs*, 6 *CmMAPKKs* and 64 *CmMAPKKKs* in melon genome. Based on structural characteristics and a comparison of phylogenetic relationships of MAPK gene families from *Arabidopsis*, cucumber and watermelon, CmMAPKs and CmMAPKKs were categorized into 4 groups, and CmMAPKKKs were categorized into 3 groups. Furthermore, chromosome location revealed an unevenly distribution on chromosomes of MAPK cascade genes in melon, respectively. Eventually, qRT-PCR analysis showed that all 14 *CmMAPKs* had different expression patterns under drought, salt, salicylic acid (SA), methyl jasmonate (MeJA), red light (RL), and *Podosphaera xanthii* (*P. xanthii*) treatments. Overall, the expression levels of *CmMAPK3* and *CmMAPK7* under different treatments were higher than those in control. Our study provides an important basis for future functional verification of MAPK genes in regulating responses to stress and signal substance in melon.

## Introduction

Plants often suffer from various stresses including abiotic stresses (low temperature, drought, salt, etc.) and biotic stresses (insects, weeds, diseases, etc.) which attenuate plant growth, development, and adversely affect the quality and productivity. It is necessary to consider the damage and response mechanisms of plants under stresses for improving resistance [1]. Signal

**Funding:** This research was supported by the National Natural Science Funding (31872131) and the Agriculture Research System of China (CARS-25).

**Competing interests:** The authors have declared that no competing interests exist.

transduction pathway is proven to be important in plant growth and stress resistance [2–4]. MAPK cascade is widely-existing and highly conserved signal transduction pathway in eukaryotes involved in plant cell differentiation [5,6], development [7,8], maturation, hormonal signal transduction [9–11], immunization and other processes [12–14] and a variety of biotic and abiotic stresses [15–18]. Therefore, MAPK cascade becomes one of the hot topics in plant research area in recent years.

MAPK cascade contains three core components: mitogen-active protein kinase kinase kinase (MAPKKK, MAP3K or MEKK), mitogen-active protein kinase kinase (MAPKK, MAP2K, MKK or MEK) and mitogen-active protein kinase (MAPK or MPK) [19]. MAPKKKs phosphorylate downstream MAPKKs through activating phosphorylation of a serine/threonine residue in the activated loop S/T-xxxxx-S/T (S: serine; X: any amino acid; T: threonine) motif contained in MAPKKs [20]. MAPKKs are a class of protein kinases with dual-specificity located at the central position of the MAPK cascade activating MAPKs by phosphorylation of tyrosine/threonine residues in the TXY (T: threonine; X: any amino acid; Y: tyrosine) motif in the MAPK activation loop [17]. MAPKs are located at the downstream of the MAPK cascade that phosphorylates a variety of substrates. Besides, MAPK structure has 11 relatively conserved subdomains (I-XI), which are all crucial for the serine/threonine protein kinase to exert its catalytic action [21–23].

MAPK cascade has been demonstrated to play crucial roles in a variety of stress and signal substance responses. Many members of the MAPK gene family from various plant species have been identified and characterized. For instance, *Arabidopsis* contains 80 MAPKKKs, 10 MAPKKs and 20 MAPKs [24, 25] which are involved in the regulation of low temperature, drought, disease resistance and other processes [12, 26–29]. The latest studies have shown that MAPKKK17/18-MKK3-MAPK1/2/7/14 participates in the signal transduction induced by abscisic acid (ABA) [30, 31]. MAPKKK3/5-MKK4/5-MAPK3/6 is involved in a variety of pathways in plant defense pathogens, including the synthesis of ethylene and phytoalexin, the thioglycoside biosynthetic pathway and stomatal immunization [32–35]. Besides, 79 MAPK cascade genes are detected in cucumber, and most of them could be induced by biotic and abiotic stresses interacting with hormones (ABA and jasmonic acid (JA)) [36]. Furthermore, 6 *ClMAPKKs* and 15 *ClMAPKs* have been identified in watermelon, and *ClMAPK7* has been found to respond to drought, heat, low temperature and salt stresses by analyzing the expression patterns of 15 *ClMAPK* genes under different stresses [37]. In cotton, *GhMAP3K15-GhMKK4-GhMAPK6-GhWRKY59-GhDREB2* participates in regulating drought response [38], while MKK4-MAPK20-WRKY40 plays a negative regulatory role in resistance to *Fusarium wilt* [39]. Identification of MAPK cascade genes from plants is therefore a necessary step in preliminary prediction of gene function in regulating plant responses. However, no systematic and genome-wide investigation of the MAPK cascade family has been carried out in melon.

Melon (*Cucumis melo* L.) as an important economic crop is widely grown in the world, and always suffered various stresses in the process of its growth and development. It has been of great significance to the study of MAPK cascade in stress resistance mechanism. In this study, we performed a comprehensive analysis of 14 *CmMAPKs*, 6 *CmMAPKKs* and 64 *CmMAPKKKs*, including their phylogenetic relationships, gene structures, conserved domains and motifs, and chromosomal locations. Then we analyzed the expression profiles of *CmMAPKs* under drought, salt, salicylic acid (SA), methyl jasmonate (MeJA), red light (RL), and *Podosphaera xanthii* (*P. xanthii*) treatments. These data will facilitate future studies to elucidate the exact biological functions of MAPK genes, and serve as a theoretical basis for genetic engineering variety improvement technology and adversity regulation in melon.

## Materials and methods

### Identification of MAPK cascade gene family members in melon

The method used to identify all the putative MAPK, MAPKK and MAPKKK family genes was similar to that used for cucumber and watermelon [36,37]. Firstly, reported amino acid sequences of MAPK, MAPKK and MAPKKK family members in *Arabidopsis*, cucumber and watermelon were downloaded from the database (*Arabidopsis* http://www.arabidopsis.org/, Cucumber http://cucurbitgenomics.org/organism/20, Watermelon http://cucurbitgenomics.org/organism/21). Then, these amino acid sequences were utilized to search against the melon database (http://melonomics.net/) using the BlastP program with an e-value of 1e$^{-10}$ as the threshold to get candidate MAPK cascade gene family members. A further search was performed with Hidden Markov Model (HMM) analysis using HMMER 3.0 program. The HMMER hits containing the serine/threonine-protein kinase-like domain (PF00069) were compared with blast results and parsed by manual editing. All data were checked for redundancy by self-BLAST and no any alternative splice variants were considered. Finally, the online software Conserved Domain Database of NCBI (http://blast.ncbi.nlm.nih.gov) and SMART database (http://smart.embl-heidelberg.de/) were used to further confirm the predicted MAPK cascade genes.

### Characteristics, conserved domain detection and subcellular location predication

ExPASy database (http://web.expasy.org/compute_pi/) was served to predict the molecular weight (MW) and isoelectric points (pI) of the melon MAPK, MAPKK and MAPKKK family proteins. MEME software (http://meme-suite.org/tools/meme) was used to analyze the conserved protein motif. The subcellular localization of MAPK cascade gene family members was analyzed using the TargetP software of the CELLO v2.5 server (http://cello.life.nctu.edu.tw/) [40].

### Multiple sequence alignments and phylogenetic construction

Multiple sequence alignments were generated using DNAMAN 9.0 software. Phylogenetic analysis was carried based on the full-length protein sequences using the MEGA 6 program by the neighbor-joining (NJ) method and a bootstrap test was carried out with 1000 interactions [41]. Built on the nomenclature, the MAPK, MAPKK and MAPKKK genes in melon were named according to the homology of the corresponding genes in *Arabidopsis*, cucumber and watermelon.

### Chromosomal location and gene structure analysis

The nucleotide sequences and chromosomal locations of all these genes were obtained from the Cucurbit Genomics Database (http://cucurbitgenomics.org/) and mapped to the chromosomes with MapInspect software, as well as the gene structures of the melon MAPK, MAPKK and MAPKKK family proteins were generated with the GSDS 2.0 (http://gsds.cbi.pku.edu.cn/).

### Plant materials and treatments

Oriental melon (*Cucumis melo* var. *makuwa* Makino) cultivar 'Yumeiren (YMR)' were grown in pots (soil: peat: compost = 1:1:1) or Yamasaki melon nutrient solution (half strength) [42] at 25˚C /18˚C under 14 h/10 h (light/dark) in a greenhouse during the whole experiments. The treatments of the seedlings were shown as follows.

For drought and salt stress treatments, seedlings at two-leaf stage grown in pots were transferred to Yamasaki melon nutrient solution and refreshed every 2 days. Then 8% (w/v) PEG-6000 and 100 mM NaCl were added to simulate drought and salt stress when seedlings were four fully expanded leaves, respectively. For SA and MeJA treatments, seedlings at four-leaf stage were sprayed with 5 mM SA and 100 μM MeJA (10 mL per plant) or with 0.1% ethanol as a control, respectively. Leaves were collected at 0 h, 1 h, 3 h, 24 h, 120 h after treatments for qRT-PCR analysis. For RL treatment, leaves were sampled after exposed to RL (160 μmol m$^{-2}$ s$^{-1}$) and white light (WL, 160 μmol m$^{-2}$ s$^{-1}$) for 6h. For biotic stress, *P. xanthii* were collected from the infected plants in the field and evenly distributed on the third and fourth leaves of the melon seedlings. In the meantime, RL were used as the inducer of the defense response in the *P. xanthii* infection experiment and untreated seedlings were taken as control. Leaves were collected after treatments at 0 d, 1 d, 3 d, 5 d, 7 d and 9 d for qRT-PCR analysis.

### RNA isolation, cDNA synthesis and qRT-PCR

The ultrapure RNA kit (Cat.CW0581, Beijing, China) was used to extract total RNA from leaves according to the manufacturer's instructions. cDNA was synthesized using PrimeScript™ RT Master Mix (Perfect Real Time) (TAKARA, Japan) and the relative expression of the *CmMAPK* genes was quantified by qRT-PCR according to the SYBR® Premix Ex Taq™ (Tli RNaseH Plus) (TAKARA, Japan) manufacturer's protocol on an ABI PRISM 7500 real-time fluorescent quantitative PCR (Applied Biosystems, Thermo Fisher Scientific, USA). The primers for the qRT-PCR analysis were designed to avoid the conserved regions and listed in the S2 Table [43]. The 2$^{-\Delta\Delta Ct}$ method was used to measure relative expressions of genes, and *18s rRNA* from melon was used as the internal control gene. All experiments were performed with three biological and three technical replicates.

### Statistical analysis

The data were analyzed using the SPSS 22.0 statistics program, and significant differences were compared using a one-way ANOVA following Duncan's multiple range test at $p < 0.05$ level except expression profiles of *CmMAPKs* under RL induction were conducted by t test at $p < 0.05$ level.

## Results

### Identification of MAPK cascade genes in melon

To identify MAPK, MAPKK and MAPKKK family genes, we conducted respective BlastP searches against the melon protein databases using query protein sequences from *Arabidopsis*, cucumber and watermelon. Meanwhile, HMM search was used to identify the serine/threonine-protein kinase-like domains (PF00069) in all potential MAPK cascade sequences from melon. After removing redundancies, alternative splices, multiple steps of screening and validation of the conserved domains, we finally identified 14 *CmMAPK* genes, 6 *CmMAPKK* genes, and 64 *CmMAPKKK* genes. The sequence data of all above MAPK cascade genes was downloaded from the Cucurbit Genomics Database (http://cucurbitgenomics.org/ search/ genome/18). Each gene was named according to its homology with *Arabidopsis* MAPK, MAPKK, or MAPKKK proteins [44].

As shown in Table 1, 14 *CmMAPKs* were randomly distributed on 8 chromosomes. These *CmMAPKs* were predicted to encoding 370 (*CmMAPK3*, *CmMAPK4-2*, *CmMAPK13*) to 645 (*CmMAPK9-1*) amino acids (aa) in length, with putative MW ranging from 42.6 (*CmMAPK13*) to 73.1 kDa (*CmMAPK9-1*) and theoretical pIs ranging from 5.01 (*CmMAPK13*) to 9.28

**Table 1. Summary of *CmMAPK* and *CmMAPKK* genes in melon.**

| Gene family | Genes | protein length(aa) | MW (KDa) | pI | Gene ID | Chromosome | Location | Direction | Subcellular location |
|---|---|---|---|---|---|---|---|---|---|
| MAPK | *CmMAPK1* | 386 | 44.7 | 6.44 | MELO3C010966 | 3 | 27457945.. 27461816 | - | Nuclear |
| | *CmMAPK3* | 370 | 42.7 | 5.46 | MELO3C020718 | 12 | 3101639.. 3105517 | - | Nuclear/ Cytoplasm |
| | *CmMAPK4-1* | 383 | 44.0 | 6.09 | MELO3C005705 | 9 | 22486536.. 22490938 | - | Cytoplasm/Nuclear |
| | *CmMAPK4-2* | 370 | 42.9 | 5.97 | MELO3C021187 | 11 | 28603135.. 28607749 | - | Nuclear/ Cytoplasm |
| | *CmMAPK6-1* | 405 | 46.3 | 5.44 | MELO3C011444 | 3 | 23672755.. 23684497 | + | Cytoplasm |
| | *CmMAPK7* | 371 | 42.7 | 6.57 | MELO3C016399 | 7 | 22631960.. 22634413 | + | Mitochondrial/ Nuclear/Cytoplasm |
| | *CmMAPK9-1* | 645 | 73.1 | 7.34 | MELO3C017350 | 2 | 23214920.. 23220783 | + | Nuclear |
| | *CmMAPK9-2* | 469 | 54.0 | 6.82 | MELO3C018306 | 10 | 19624772.. 19630739 | - | Cytoplasm/Nuclear |
| | *CmMAPK9-4* | 479 | 54.8 | 8.56 | MELO3C002124 | 12 | 25003151.. 25007636 | + | Nuclear/ Cytoplasm |
| | *CmMAPK13* | 370 | 42.6 | 5.01 | MELO3C026848 | 12 | 3463126.. 3466823 | + | Nuclear |
| | *CmMAPK16* | 565 | 64.5 | 8.81 | MELO3C021394 | 11 | 26907460.. 26913601 | + | Cytoplasm/Nuclear |
| | *CmMAPK19* | 492 | 56.6 | 9.22 | MELO3C016139 | 7 | 19589085.. 19593842 | + | Cytoplasm/Nuclear |
| | *CmMAPK20-1* | 606 | 68.3 | 9.10 | MELO3C014472 | 5 | 2160298.. 2164803 | + | Nuclear |
| | *CmMAPK20-2* | 620 | 70.4 | 9.28 | MELO3C019687 | 11 | 23069620.. 23074916 | + | Nuclear/Mitochondrial |
| MAPKK | *CmMAPKK2-1* | 300 | 34.2 | 5.38 | MELO3C004635 | 5 | 28117295.. 28119504 | - | Cytoplasm |
| | *CmMAPKK2-2* | 340 | 38.3 | 5.11 | MELO3C015497 | 2 | 2320452.. 2323356 | - | Cytoplasm |
| | *CmMAPKK3* | 518 | 57.8 | 5.52 | MELO3C026577 | 4 | 26205457.. 26210900 | - | Cytoplasm |
| | *CmMAPKK5* | 368 | 41.4 | 9.04 | MELO3C025790 | 4 | 18804288.. 18805851 | + | Nuclear |
| | *CmMAPKK6* | 360 | 40.6 | 6.60 | MELO3C004594 | 5 | 27851752.. 27855233 | - | Cytoplasm/Nuclear |
| | *CmMAPKK9* | 321 | 36.0 | 8.25 | MELO3C002150 | 12 | 24839711.. 24840980 | + | Nuclear |

(*CmMAPK20-2*). The subcellular localization was predicated and found that *CmMAPKs* were located in nuclear and cytoplasm, except *CmMAPK7* and *CmMAPK20-2* may be present in mitochondria. Meanwhile, 6 *CmMAPKKs* were predicted to encoding 340–518 aa, with MW ranging from 38–57.8 kDa and pIs ranging from 5.11–9.04. They were predicted to be located in cytoplasm and nuclear. All 64 *CmMAPKKK*s were randomly distributed on 13 chromosomes. The polypeptide lengths of these MAPKKKs ranged from 221 to 1208 aa with MW ranging from 24.6 to 133.5 kDa and pIs ranging from 4.63 to 9.41. Besides, 47 of the *CmMAPKKK*s were located in nuclear and cytoplasm, while the others may be present in plasma membrane, mitochondria or chloroplast (S1 Table).

## Phylogenetic relationship and structures of MAPK cascade genes

To further characterize the MAPK cascade genes, unrooted phylogenetic trees were produced by aligning the full-length protein sequences of all MAPK cascade genes in *Arabidopsis*, watermelon, cucumber and melon using NJ method.

Phylogenetic analysis with *Arabidopsis*, watermelon, cucumber and melon revealed that 14 CmMAPKs were divided into 4 groups (A, B, C, and D) (Fig 1). Two members were assigned to group A (CmMAPK3 and CmMAPK6-1), 3 members to group B (CmMAPK4-1, CmMAPK4-2 and CmMAPK13), 2 members to group C (CmMAPK1 and CmMAPK7), and the remaining 7 members were assigned to group D (CmMAPK9-1, CmMAPK9-2, CmMAPK9-4, CmMAPK16, CmMAPK19, CmMAPK20-1 and CmMAPK20-2). Analysis of gene structures showed that 5 members of group A and B contained 6 introns, the members of group C contained 2 introns, MAPK members in group D had a relatively high number of introns, generally 9-11(Fig 2). Likewise, CmMAPKKs were parted into 4 groups, named A, B,

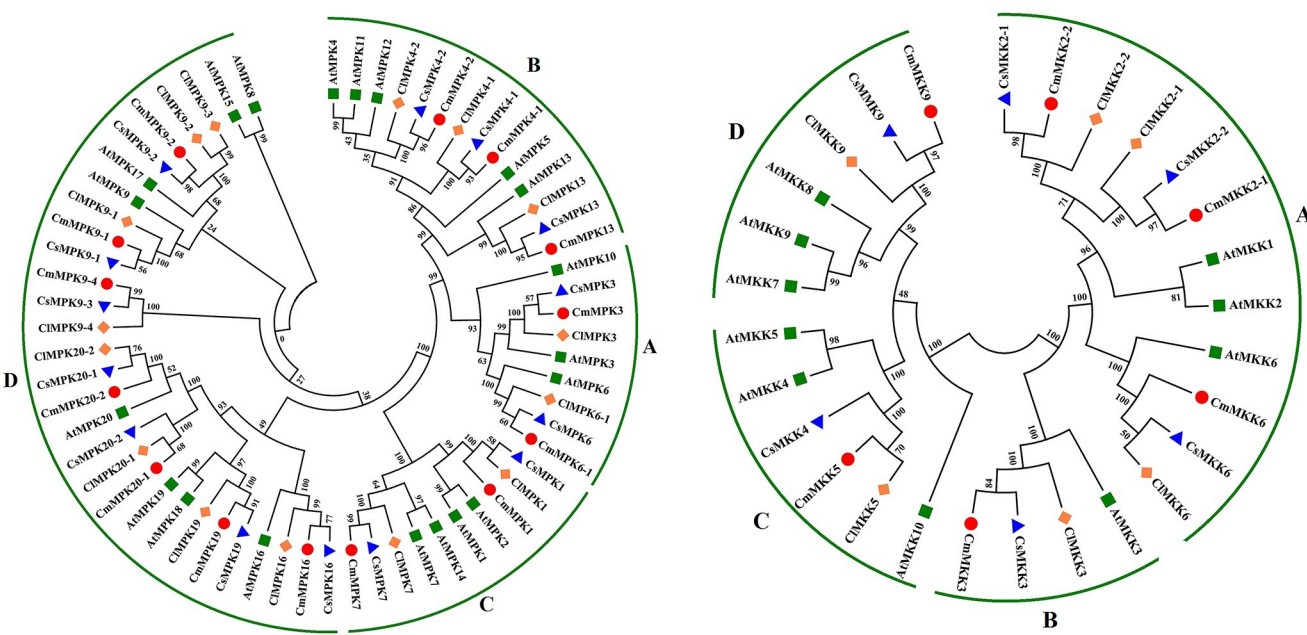

**Fig 1. Phylogenetic relationships of Arabidopsis, cucumber, watermelon and melon MAPK and MAPKK proteins.** The unrooted tree was constructed, using the MEGA6.0 program, by the NJ method. Bootstrap values were calculated for 1000 replicates.

C and D. Except group A contained three members (CmMAPKK2-1, CmMAPKK2-2 and CmMAPKK6), each of the other groups contained one member. Notably, CmMAPKK5 and CmMAPKK9 were closely related in phylogenetic development, but belonged to different groups (Fig 1). Analysis of the gene structures showed that all three members of group A contained 8 introns, *CmMAPKK3* contained 9 introns in group B, and *CmMAPKK5* and *CmMAPKK9* contained 1 intron in group C and D respectively (Fig 2).

The MAPKKK gene family in plants is usually classified into 3 categories, namely MEKK, Raf, and ZIK [45–46]. Refer to the classification method of *Arabidopsis*, MEKK group contained 20 members, Raf 34 members and ZIK 10 members (S1 Fig). Genetic structures analysis showed that the number of introns of MAPKKK gene family members in melon varied greatly from 2 to 36. MAPKKK members in the same group also had different genetic structures. In MEKK group, 9 genes contained 2 introns, 3 genes contained 4 introns, and the remaining 8 genes contained 16–36 introns. In ZIK group, 2 genes contained 4 introns, 1 gene contained 16 introns, and the remaining 7 genes contained 14 introns. RAF group members had the largest differences in gene structure among 3 groups, with the number of introns ranging from 4 to 32. Through careful analysis, it was found that MAPKKK genes had similar or identical gene structure, these genes owned high homology and close phylogenetic relationship. For example, *CmMEKK21-1*, *CmMEKK21-2*, *CmMEKK21-3*, *CmMEKK21-4* and *CmMEKK21-5* contained two introns, and *CmZIK4-1*, *CmZIK4-2* and *CmZIK4-3* contained 14 introns. Both *CmRAF1-1* and *CmRAF1-2* contained 30 introns (S2 Fig).

## Conserved domains and motifs in MAPK cascade proteins

To survey the structural domain features of MAPK cascade gene families, a multiple sequence alignment of the predicted amino acid residues was carried out using the DNAMAN 9.0 software. The results showed that all CmMAPK proteins had conserved motif TxY and CD domain (LH) DxxDE (P) xC existing in group A and B (Fig 3). The conserved motifs of the

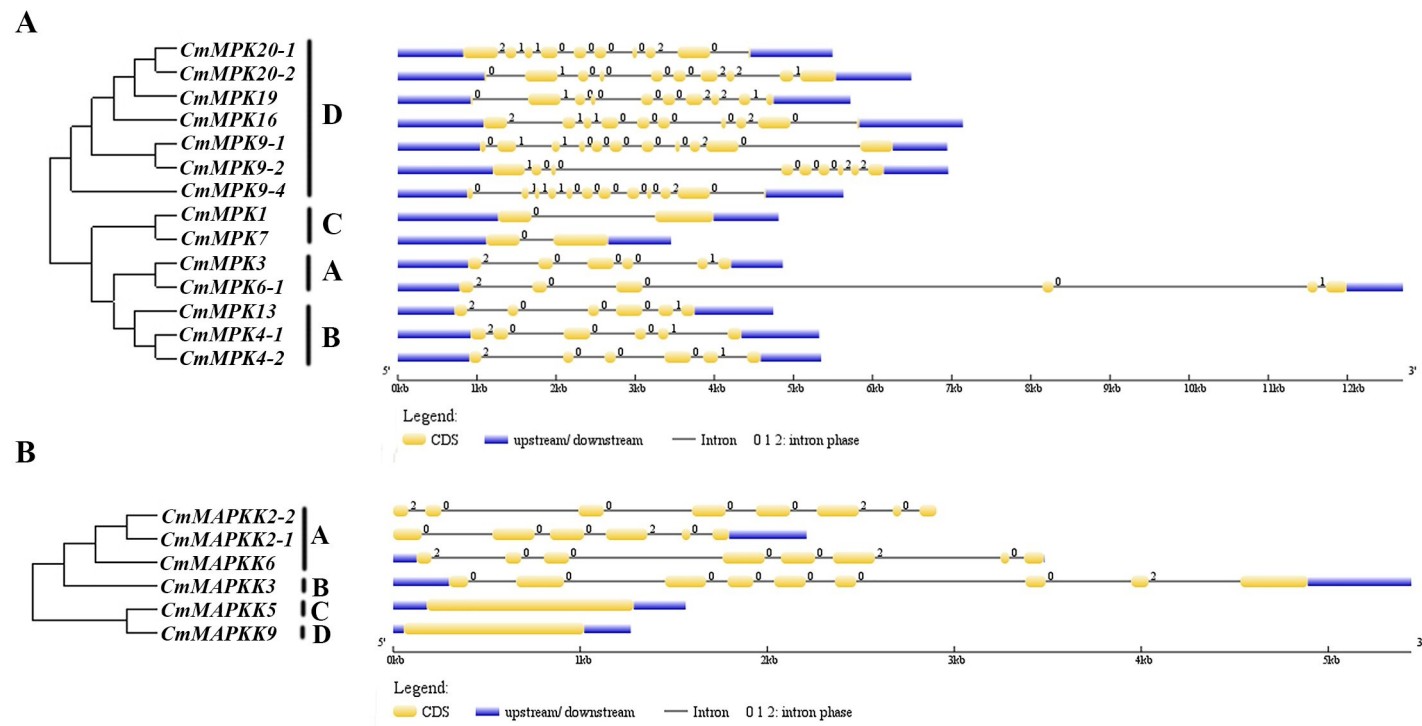

**Fig 2. Exon-intron structures of *CmMAPK* and *CmMAPKK* genes.** Exons and introns are shown as yellow boxes and thin lines respectively.

CmMAPK proteins were analyzed using the online tool MEME. Results displayed that all CmMAPK proteins contained motif 2–8, motif 1 only existed in group A and B, and motif 9–10 only existed in group C and D (Fig 4). Multiple alignment of the CmMAPKK proteins showed that all proteins contained conserved D (L/I/V) K, the phosphorylation target site S/T-xxxxx-S/T and signature VGTxxYMSPER, which were conserved in the catalytic domain (Fig 3) [30]. Motif analysis found that CmMAPKK6, CmMAPKK5, and CmMAPKK9 contained motif 1–8 and the rest had motif deletion (Fig 4). These motifs may lead to different functions of each gene, which further upheld the accuracy of classification.

Multiple sequence alignment of CmMAPKKKs from melon showed that melon and other objects were similar to MEKK group containing conserved G(T/S)Px(W/Y/F)MAPEV domain, RAF group containing conserved GTxx(W/Y)MAPE domain and ZIK group containing conserved GTPEFMAPE(L/V)Y domain (S3 Fig). MEME tool analysis found that all MAPKKKs contained motifs 1–8, motif 9 only existed in the ZIK group, and motif 10 existed in the MEKK and RAF group (S4 Fig). This result further verified the accuracy of group classification, and speculated that different gene functions of different groups in the same family may be determined by the specific motifs.

Through the analysis of protein conserved motifs and protein functional domains, it was found that members of the MAPK cascade pathway gene family in melon were very conserved in structure, therefore we can predict their functions in melon based on the reported homologous gene functions in *Arabidopsis* [44], cucumber [36] and watermelon [37].

## Chromosomal localization of MAPK cascade genes

BLASTN searches were performed against the melon genome database using the DNA sequence of each MAPK cascade gene to determine the chromosomal distribution and

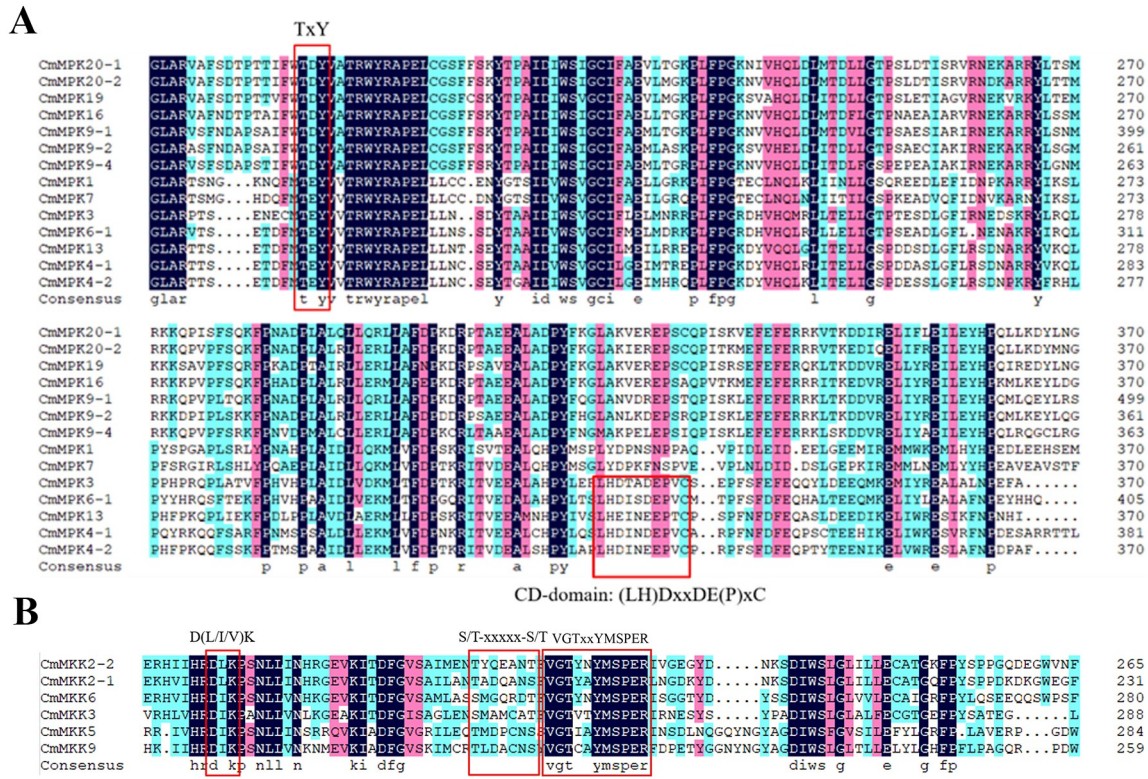

**Fig 3. Multiple sequence alignment analysis of melon MAPK and MAPKK proteins carried out using the DNAMAN 9.0 program.** Conserved domains in MAPK and MAPKK proteins are marked with a red box and indicated.

transcriptional direction of the MAPK cascade genes in melon. By chromosome localization analysis, it was found that 84 genes in MAPK cascade pathway were distributed on 13 chromosomes (Fig 5). It was notable that some genes showed high homology, but they were distributed on different chromosomes such as *CmMAPK4-1/CmMAPK4-2*, *CmMAPK9-1/ CmMAPK9-2/CmMAPK9-4*, *CmMAPK20-1/CmMAPK20-2*, *CmMKK2-1/CmMKK2-2* and *CmZIK4-1/CmZIK4-2/CmZIK4-3*. From the above results, we could infer that there were a large number of gene duplications and gene transfer events in the long-term evolution of members of the MAPK cascade pathway gene family in melon. They participated in both chromosome fragment replication events and tandem replication events, which were of profound significance for the expansion of MAPK cascade pathway gene family members in melon.

## Expression profiles of *CmMAPKs* in melon seedlings under drought and salt stress

In open field cultivated of melon plants, it is often affected by drought and soil salinization, which further affects the yield and quality of melon. In order to study *CmMAPKs* expression in melon under drought and salt stress, we used nutrient solution cultivation way, by adding 8% PEG into the nutrient solution (simulation of drought stress) and 100 mM NaCl (simulation of salt stress) to treat melon seedlings. It was found that the expression pattern of *CmMAPK* genes in melon changed, and the expression trend of multiple genes was consistent under the same treatment (Fig 6). Under drought stress, except that the transcription levels of *CmMAPK19*, *CmMAPK20-1* and *CmMAPK20-2* decreased, the other members increased at 6 h, 12 h and 24 h after treatment, and then decreased to the original level. These results

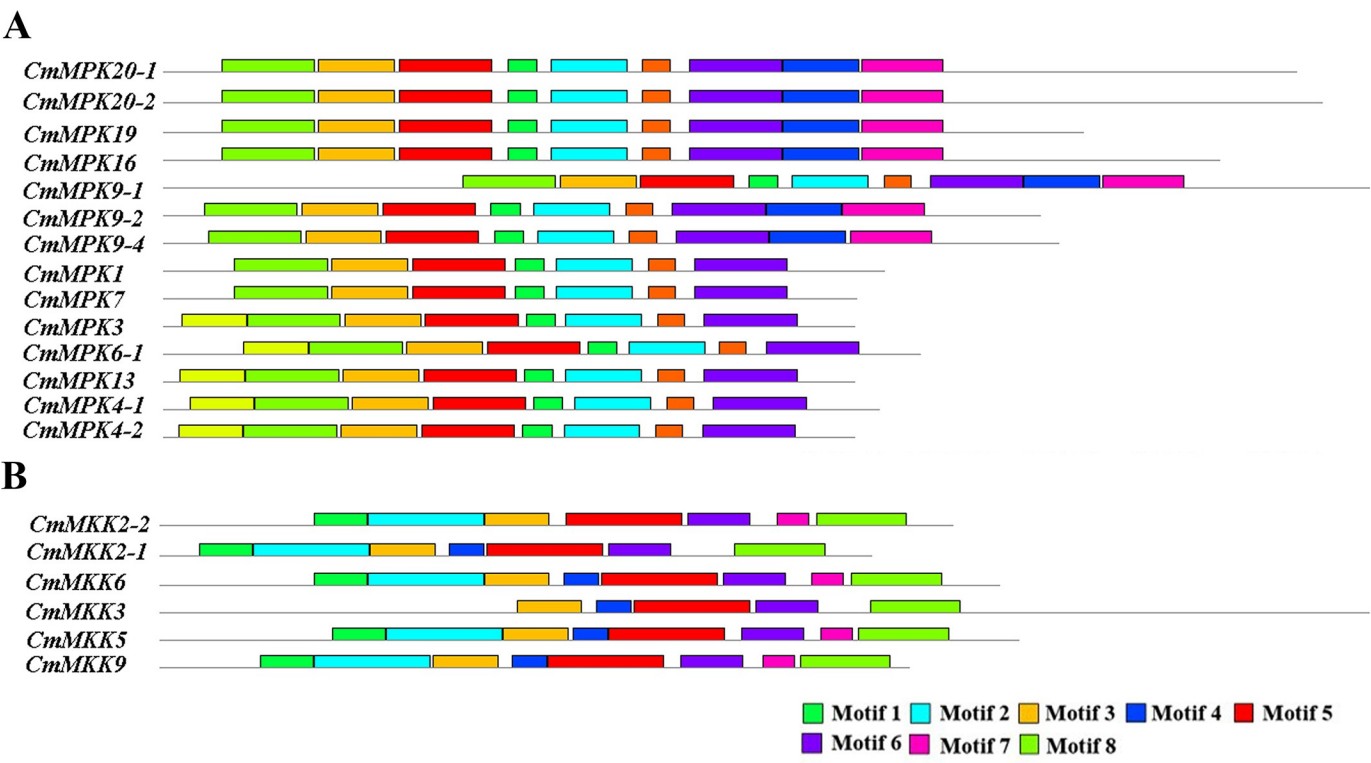

**Fig 4. Conserved motifs in MAPK and MAPKK proteins.** Motifs were determined using MEME program. Grey lines represent the non-conserved sequence, and each motif is indicated by a colored box numbered at the bottom.

indicated that these up-regulated genes might be critical genes respond to the drought stress. Under salt stress, only *CmMAPK3*, *CmMAPK7*, *CmMAPK20-1* and *CmMAPK20-2*

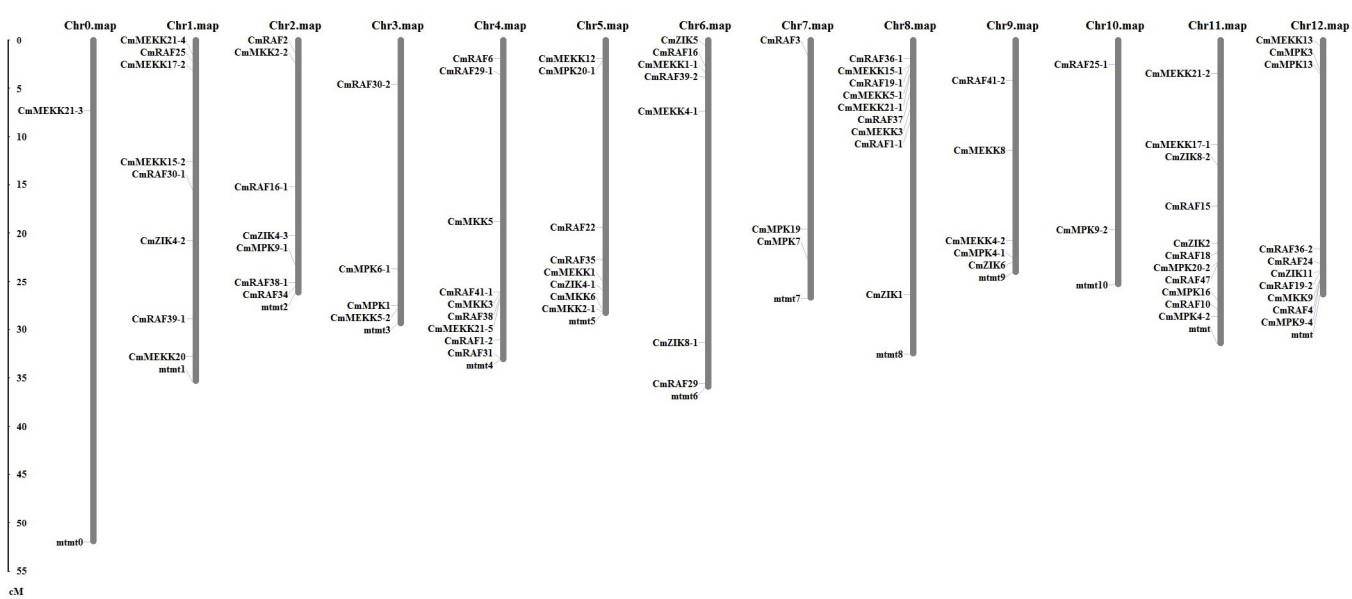

**Fig 5. Distribution of MAPK cascade genes on melon chromosomes according to the chromosome map.** In total, 84 genes were mapped to 13 chromosomes. The scale is in centiMorgans (cM).

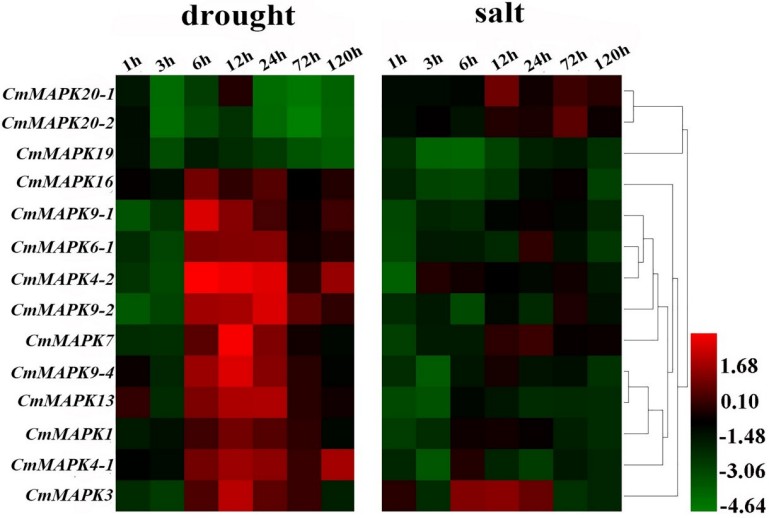

**Fig 6. Expression analysis of 14 *CmMAPK* genes in leaves under drought and salt stresses.** $Log_2$ based values (signal values for treatment/signal value for control) were used to create the heat map based on transcriptomic data for *CmMAPKs*. The color scale (representing signal values) is shown at the bottom.

significantly induced at 6 h, 12 h or 24 h after treatment, indicating that the four genes may play an important role in response to salt stress.

## Expression profiles of *CmMAPKs* in response to SA and MeJA treatment

In plant cells, hormones can affect physiological and biochemical reactions of plants through a variety of signal transduction pathways. SA and JA are important hormones in plant immune system. To study whether *CmMAPKs* in melon were regulated by hormone signaling sub-stances, exogenous spraying of 5 mM SA and 100 μM MeJA on leaves of melon seedlings was carried out in this study. Cluster analysis was performed on the data results. After SA induc-tion, except the down-regulated expression of *CmMAPK20-1* and *CmMAPK20-2*, other genes induced to up-regulate expression, indicating that different members of *CmMAPKs* played dif-ferent roles in SA-induced defense response. At the later stage of MeJA induction, all 14 *CmMAPKs* strongly induced, revealing that members of *CmMAPKs* positively responded to all the induced defense responses of MeJA and played an essential role in this process (Fig 7). In summary, members of *CmMAPKs* may be widely involved in signaling hormone-induced defense responses and play different roles.

## Expression profiles of *CmMAPKs* in melon seedlings under RL induction

Studies have shown that moderate red light induction can improve the resistance of plants to Pst-DC3000 [47]. Our preliminary study of the research group showed that the disease resis-tance significantly increased by pre-irradiation of red light (160 μmol m$^{-2}$ s$^{-1}$) for 6h at melon seedling stage, when melon seedlings were inoculated with *P. xanthii*. In order to explore whether *CmMAPKs* were involved in the defense reaction induced by red light, this experi-ment conducted red light irradiation on melon seedlings and white light irradiation as the con-trol. The results showed after red light induction, the transcription levels of *CmMAPK3*, *CmMAPK6-1* and *CmMAPK7* in melon leaves remarkably increased compared to the control. It was worth noting that the transcription levels of *CmMAPK3* were up to 5 folds higher than that of the control, suggesting that they might play a positive regulatory role in the red light

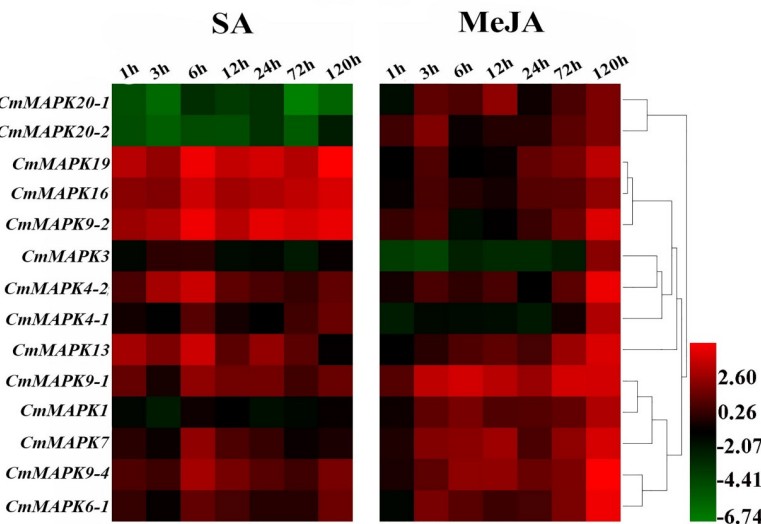

**Fig 7. Expression analysis of 14 *CmMAPK* genes in leaves in response to SA and MeJA.** $Log_2$ based values (signal values for treatment/signal value for control) were used to create the heat map based on transcriptomic data for *CmMAPKs*. The color scale (representing signal values) is shown at the bottom.

induced defense response. However, the transcription levels of *CmMAPK20-1* and *CmMAPK20-2* were not significantly different indicating that they were not sensitive to red light induction and played an auxiliary role in the red light induced defense response. The transcription levels of other gene family members significantly down-regulated after red light induction, which may act as a negative regulatory role in the red light induced defense response (Fig 8).

## Expression profiles of *CmMAPKs* in response to pathogen infection

Gene expression patterns are typically used as an indicator of gene function. In the protected cultivation of melon, high temperature and high humidity environment make it vulnerable to the pathogen of *P. xanthii*, which restricts the growth of melon and seriously affects the yield and quality. Therefore, we selected *P. xanthii* as the biotic stress factor to infect the leaves of melon in this study. Cluster analysis was performed on the data results (Fig 9). In addition to *CmMAPK 1*, *CmMAPK6-1*, *CmMAPK7* and *CmMAPK 19*, other *CmMAPKs* greatly up-regulated on the third day after infection by *P. xanthii*. Moreover, the expression level of these *CmMAPKs* gradually decreased with the increase of the invasion time of *P. xanthii*, except *CmMAPK3*, *CmMAPK4-2* and *CmMAPK13*. At the same time, the transcription levels of most *CmMAPKs* with pre-irradiation of RL were significantly lower than those of *P. xanthii* inoculated alone at 3 and 5 days, and were significantly upregulated at 7 and 9 days after inoculation. It is speculated that *CmMAPKs* may play a negative regulatory role in the mechanism of disease resistance after irradiation of red light.

## Discussion

During plant growth and development, it is inevitable to be subjected to stresses caused by external environments affecting its growth and development, while plants will form a series of defense reaction mechanisms. Reversible phosphorylation of proteins plays an important part in the recognition, gradual transduction and amplification of extracellular signals. Mitogen-activated protein kinase (MAPK), a type of serine/threonine protein kinase activated by the

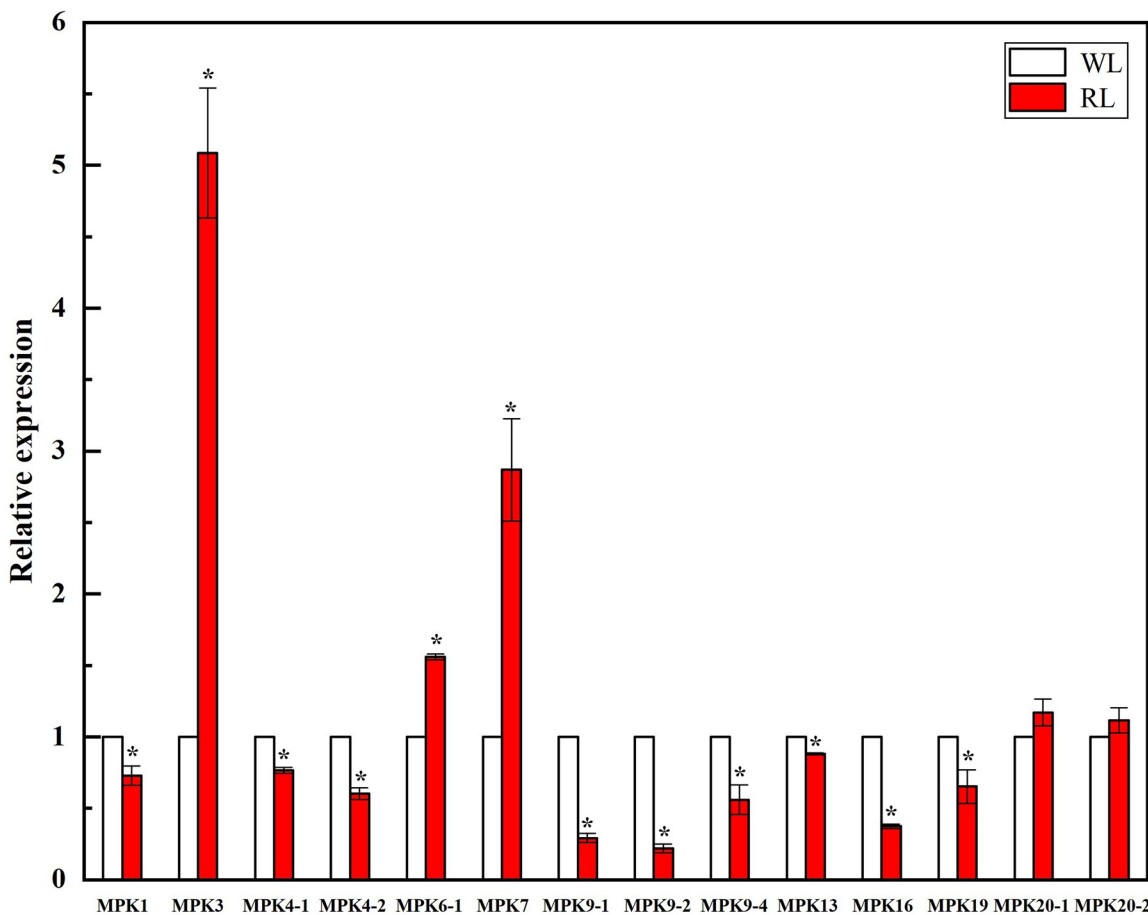

**Fig 8. Expression analysis of *CmMAPK* genes in leaves in response to RL, carried out by qRT-PCR.** The *18s rRNA* was used as an internal control. The experiment included three biological replicates, each with three technical replicates. Values represent means of n = 9±SD (t test: *, P < 0.05). WL, white light; RL, red light.

MAPK cascade (MAPKKK-MAPKK-MAPK), transmits the upstream signal cascade to the downstream response molecule via protein phosphorylation, thereby activating the expression of the stress-resistant genes, making the plant adapt to the adversity [48]. However, little is known about the identity and functions of melon MAPK cascade genes. Thus, we mapped out the major structural characteristics of all MAPK cascade genes and expression profiles of *CmMAPKs* in melon.

In the present study, a comprehensive set of 84 MAPK cascade genes were identified including 14 *CmMAPKs*, 6 *CmMAPKKs* and 64 *CmMAPKKKs* through BlastP homologous alignment combined with HMMER searches. Analyzing the amino acid sequences of MAPK gene members from *Arabidopsis* and other species, MAPK family members are divided into 4 groups: A, B, C, and D. Among them, MAPK proteins in A, B, and C contain TEY phosphorylation sites, while proteins in D contains TDY phosphorylation sites [44]. In our investigation of MAPK cascade, phylogenetic trees suggested that melon CmMAPK and CmMAPKK were mainly categorized into 4 groups, and CmMAPKKK were categorized into 3 groups, revealing the evolutionary homology between melon and *Arabidopsis*. Moreover, we found that the melon MAPK cascade genes are structurally conservative by analyzing the protein conserved motifs and protein functional domains. For example, CmMAPK and CmMAPKK proteins contained the conserved motif TxY and S/T-xxxxx-S/T, respectively. Previous research has

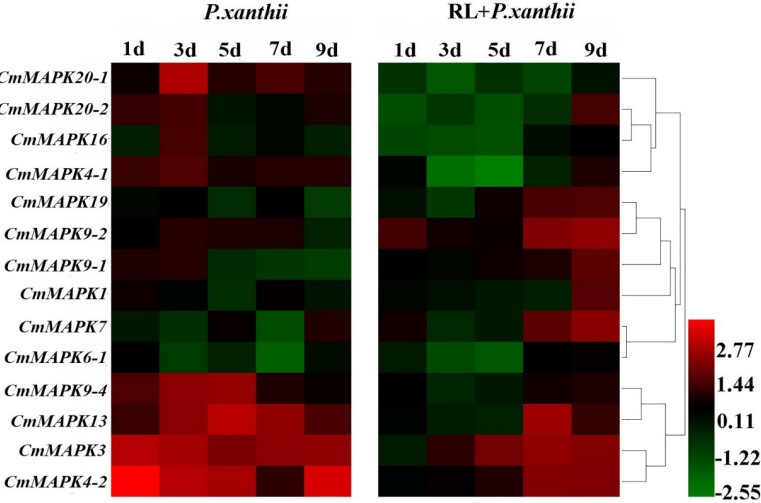

**Fig 9. Expression analysis of 14 *CmMAPK* genes in leaves under *P. xanthii* and RL+ *P. xanthii* stresses.** Log$_2$ based values (signal values for treatment/signal value for control) were used to create the heat map based on transcriptomic data for *CmMAPKs*. The color scale (representing signal values) is shown at the bottom. *P. xanthii*, inoculated with *Podosphaera xanthii* alone; RL+ *P. xanthii*, inoculated with *Podosphaera xanthii* after irradiation of red light.

suggested that gene duplication events play an important role in the expansion of gene families [49]. In our study, chromosome localization showed that 84 members of the MAPK cascade gene family of melon were distributed on 13 chromosomes indicating that these genes were involved in chromosome fragment replication events and tandem replication, which are indispensable for the expansion of family members. These features in melon have also been observed in *Arabidopsis* [44], cucumber [36] and watermelon [37], which supports our identification of MAPK cascade genes.

Stress resistance and disease resistance are the result of mutual adaptation and selection between plants and environmental stresses and pathogens. The expression of gene changes with environmental stress and pathogen infection. It is previously shown that *Arabidopsis* MAPK1/2/7/14 are induced by ABA [30, 31]. Some of watermelon *ClMAPKs* (*ClMAPK1*, *ClMAPK3*, *ClMAPK4-1*, *ClMAPK4-2*, *ClMAPK6*, *ClMAPK7*, *ClMAPK9-1*, *ClMAPK9-2*, *ClMAPK13*, *ClMAPK16*, *ClMAPK19*, *ClMAPK20-1*, *ClMAPK20-2*) can respond to drought stress [37]. However, there is no information available about the response to *CmMAPKs* in stress-responsiveness. In our study, the transcripts of the 14 genes were firstly detected in leaves under drought, salt, SA and MeJA, RL and *P. xanthii* treatments using qRT-PCR. As shown from the heat map in Fig 6, some genes like *CmMAPK19*, *CmMAPK20-1* and *CmMAPK20-2* were down-regulated, and the remaining *CmMAPKs* decreased firstly after treatment and then returned to normal levels under drought stress. Besides, *CmMAPK3*, *CmMAPK7*, *CmMAPK20-1* and *CmMAPK20-2* significantly up-regulated under salt stress. After SA treatment, the expression levels of *CmMAPK20-1* and *CmMAPK20-2* were notably lower than that of the control, and other members were up-regulated. Nevertheless, *CmMAPKs* were strongly induced to express after spraying with MeJA for 120 h (Fig 7). After treatment with RL, *CmMAPK3*, *CmMAPK6-1* and *CmMAPK7* were up-regulated, even the expression of *CmMAPK3* was up to 5-fold compared with control, but the expression levels of *CmMAPK20-1* and *CmMAPK20-2* changed little and other members were down-regulated (Fig 8). In addition, *CmMAPK1*, *CmMAPK6-1* and *CmMAPK19* were down-regulated, and the remaining members were expressed in different degrees after inoculation with *P. xanthii*.

Interestingly, after pretreatment with RL, the expression of *CmMAPKs* was lower than that of *P. xanthii* inoculated alone, while some *CmMAPKs* were up-regulated, such as *CmMAPK3*, *CmMAPK4*, *CmMAPK7* and *CmMAPK9-2*, suggesting that *CmMAPKs* may play a negative regulatory role in the red light-induced resistance to *P. xanthii* (Fig 9). The results suggested that *CmMAPK3* and *CmMAPK7* were up-regulated remarkably after different treatments indicating that these two genes may participate in multiple signal transduction pathways at the same time, playing important roles in the melon resistance network. It is found that *Arabidopsis* MAPK3 belongs to group A. The protoplasts are treated with bacterial flagellin (flg22) to detect MAPK activity: MAPK3 is activated, and the downstream transcription factor WRKY22/29/33 are activated to increase plant resistance [12, 28, 29]. Previous studies have proven that *AtMAPK3* is involved in the synthesis of ethylene and phytoalexin, the biosynthesis of indole glucosinolates and stomatal immunity [32–35]. Besides, MAPK3/6 negatively regulates the tolerance of *Arabidopsis* to low temperature by phosphorylating ICE1 [50]. Further, on the 9th day after infection of *Fusarium oxysporum*, the expression of *ClMAPK3* is significantly higher than that of the control [37]. Thus, MAPK3 is widely involved in resistance to stresses in plants. The MAPK members of different group are involved in many signal transductions while there are less reports and gene function studies of group C [51, 52]. We found that CmMAPK1 and CmMAPK7 identified in melon belonged to group C of the MAPK gene family. In *Arabidopsis*, AtMAPK1, AtMAPK2, AtMAPK7, and AtMAPK14 are classified as group C, of which AtMAPK7 may participate in pathogen signaling pathways that regulate PR gene expression [53]. *GhMAPK7* has been documented to take part in plant disease resistance, which expression can be activated by pathogen infection and multiple defense response signaling molecules [54]. In addition, *ZmMAPK7* can improve the tolerance to oxidative stress and found that it mediates ABA and $H_2O_2$ signaling processes in transgenic tobacco [55]. The expression patterns of watermelon MAPK family members in abiotic stress, hormone induction and *Fusarium oxysporum* infection show that the overlapping expression characteristics of *ClMAPK7* gene to various stress factors predicted the pleiotropic effect of *ClMAPK7* [37]. Our findings suggested that *CmMAPK3* and *CmMAPK7* homologous to *AtMAPK3* and *AtMAPK7* were involved in various abiotic stresses, hormone signal induction, RL induction and *P. xanthii* infection. However, whether the function is consistent with that of the identified and reported genes in each of the above plants should be verified by transient expression and gene editing techniques. Besides, in-depth research of *CmMAPK7* can complement and improve the function of group C.

## Conclusions

In our study, 14 *CmMAPKs*, 6 *CmMAPKKs* and 64 *CmMAPKKKs* were identified in the genome of melon. CmMAPKs and CmMAPKKs were grouped into 4 groups (A, B, C and D) while CmMAPKKKs were grouped into 3 groups (MEKK, ZIK and RAF). Besides, the study provided the first comprehensive evaluation of the phylogenetic relationships, gene structures, conserved domains and motifs, and chromosomal localizations of MAPK cascade genes. Furthermore, we analyzed expression profiles of *CmMAPKs* under drought, salt, SA, MeJA, RL and *P. xanthii* treatments. Most of the *CmMAPK* genes were induced by biotic and abiotic stress, as well as signal substance (RL, SA and MeJA). Our results also suggested the expression levels of *CmMAPK3* and *CmMAPK7* in different treatments were higher than those in control, suggesting that they were widely involved in various stress resistance mechanisms, which could be used as future research priorities.

## Supporting information

**S1 Fig. Phylogenetic relationships of *Arabidopsis*, cucumber and melon MAPKKK proteins.** The unrooted tree was constructed, using the MEGA6.0 program, by the NJ method. Bootstrap values were calculated for 1000 replicates.
(TIF)

**S2 Fig. Exon-intron structures of *CmMAPKKK* genes.** Exons and introns are shown as yellow boxes and thin lines respectively.
(TIF)

**S3 Fig. Multiple sequence alignment analysis of melon MAPKKK proteins carried out using the DNAMAN 9.0 program.** Conserved domains in MAPKKK proteins are marked with a red box and indicated.
(TIF)

**S4 Fig. Conserved motifs in MAPKKK proteins.** Motifs were determined using MEME program. Grey lines represent the non-conserved sequence, and each motif is indicated by a colored box numbered at the bottom.
(TIF)

**S5 Fig. The related supporting information of primer design for qRT-PCR analysis.**
(TIF)

**S1 Table. Summary of *CmMAPKKK* genes in melon.**
(DOCX)

**S2 Table. Primers used in this study.**
(DOCX)

## Acknowledgments

The authors really appreciate Wei Liu, Chenghui Wang, Meng Li, Fei Luo and Qi Li for the assistance of experiment and result analysis.

## Author Contributions

**Data curation:** Yuepeng Li, Qiaojuan Xing.

**Formal analysis:** Xiaolan Zhang, Yuepeng Li, Lingqi Yue.

**Writing – original draft:** Xiaolan Zhang.

**Writing – review & editing:** Hongyan Qi.

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
