## [Decision Letter · Decision Letter 0]

6 Mar 2020

PONE-D-20-01288

Genome-wide identification of mitogen-activated protein kinase (MAPK) cascade and expression profiling of CmMAPKs in melon (Cucumis melo L.)

PLOS ONE

Dear Dr Qi,

Thank you for submitting your manuscript to PLOS ONE. After careful consideration, we feel that it has merit but does not fully meet PLOS ONE’s publication criteria as it currently stands. Therefore, we invite you to submit a revised version of the manuscript that addresses the points raised during the review process.

We would appreciate receiving your revised manuscript by Apr 20 2020 11:59PM. To enhance the reproducibility of your results, we recommend that if applicable you deposit your laboratory protocols in protocols.io, where a protocol can be assigned its own identifier (DOI) such that it can be cited independently in the future. For instructions see: http://journals.plos.org/plosone/s/submission-guidelines#loc-laboratory-protocols

We look forward to receiving your revised manuscript.

Kind regards,

Yuan Huang

Academic Editor

PLOS ONE

Journal Requirements:

3. Please upload a copy of Supporting Information Table S4 which you refer to in your text on page 8.

Reviewers' comments:

Reviewer's Responses to Questions

**Comments to the Author**

1. Is the manuscript technically sound, and do the data support the conclusions?

Reviewer #1: Partly

Reviewer #2: Partly

2. Has the statistical analysis been performed appropriately and rigorously? 

Reviewer #1: Yes

Reviewer #2: No

3. Have the authors made all data underlying the findings in their manuscript fully available?

Reviewer #1: Yes

Reviewer #2: Yes

4. Is the manuscript presented in an intelligible fashion and written in standard English?

Reviewer #1: No

Reviewer #2: Yes

5. Review Comments to the Author

Reviewer #1: This manuscript describes the different expression patterns of MAPK genes under virous of treatments. There some concerns are:

1. The authors are recommended to ask help from English native speakers.

2. Please check the website of database of cucumber and watermelon.

3. exhibit chromosomal location based on the physical position.

4. The expression pattern of genes response to different treatments is different, it may be better to separately exhibit the results of different treatments.

5. the results of Expression profiles of CmMAPKs in response to pathogen infection are not clearly presented. And the conclusion, it is speculated that CmMAPKs may play a negative regulatory role in the mechanism of disease resistance, is not precise.

6. In the Discussion section, the findings should be compared with the findings in prior studies, but findings in this study and that in in prior studies should not exhibit in separate paragraphs.

Reviewer #2: Genome-wide identification and expression profiling of MAPK genes in melon were Investigated in this study, which will provide some information to the related research community. The following problems, not limiting to these, should be taken into consideration to improve the robustness of the research and the quality of manuscript.

1. Biological replication should be adopted in the experiments.

2. In qRT-PCR analysis, several reference genes have been validated in melon, which are better than 18sRNA. The authors should use the validated reference genes.

3. Family genes have high sequence similarities and share several conserved motifs among them. As a result, specificity of primers is vital for the success of qRT-PCR analysis. The authors should give the detailed information on the method of primer design and provide the related supporting figures.

4. In the section of statistical analysis, the method used is not correct.

6. PLOS authors have the option to publish the peer review history of their article (what does this mean?). If published, this will include your full peer review and any attached files.

Reviewer #1: No

Reviewer #2: No

---

## [Author Response · Author response to Decision Letter 0]

17 Apr 2020

Journal Requirements:

 Re: Thank you for your suggestion. We ensured that our manuscript meets PLOS ONE's style requirements.

 Re: Thank you for your suggestion. We created a new ORCID iD (0000-0002-7592-8749) in Editorial Manager.

3. Please upload a copy of Supporting Information Table S4 which you refer to in your text on page 8.

Re: We are sorry for our incorrect expression. Actually, Table S4 which we refer to in our text on page 8 is Table S2. Please see in the line 153 of the manuscript revision.

Comments to the Author

Reviewer #1: This manuscript describes the different expression patterns of MAPK genes under virous of treatments. There some concerns are:

1. The authors are recommended to ask help from English native speakers.

Re: We have followed your suggestion. The English native speakers have helped us to check and revised our manuscript, please see the file of “Revised Manuscript with Track Changes”.

2. Please check the website of database of cucumber and watermelon.

Re: We have followed your suggestion and revised the website. Please see in the line 96-97 of the manuscript revision. 

3. exhibit chromosomal location based on the physical position.

Re: We have followed your suggestion and reformatted the Figure 5. The physical position was exhibited on the left in the Figure 5. Please see the revised figure in the figures file.

4. The expression pattern of genes response to different treatments is different, it may be better to separately exhibit the results of different treatments.

Re: We have followed your suggestion and reformatted the Figure 6, 7 and 9. Please see the revised figures in the figures file.

5. the results of Expression profiles of CmMAPKs in response to pathogen infection are not clearly presented. And the conclusion, it is speculated that CmMAPKs may play a negative regulatory role in the mechanism of disease resistance, is not precise.

Re: We are sorry for our inaccurate expression. We have revised the sentences. Please see in the line 337-345 of the manuscript revision.

6. In the Discussion section, the findings should be compared with the findings in prior studies, but findings in this study and that in in prior studies should not exhibit in separate paragraphs.

Re: We have followed your suggestion and organized the language representations exhibiting our findings and that in prior studies in one paragraph. Please see in the line 382-434 of the manuscript revision.

Reviewer #2: Genome-wide identification and expression profiling of MAPK genes in melon were Investigated in this study, which will provide some information to the related research community. The following problems, not limiting to these, should be taken into consideration to improve the robustness of the research and the quality of manuscript.

1. Biological replication should be adopted in the experiments.

Re: We are sorry for our incorrect expression. Three biological and three technical replicates were adopted in the experiments mentioned in our manuscript. We have revised it. Please see in the line 154-155 of the manuscript revision.

2. In qRT-PCR analysis, several reference genes have been validated in melon, which are better than 18s rRNA. The authors should use the validated reference genes.

Re: We have considered your suggestion. Please allow me to explain that the reference gene we used is more stable than actin after years of our laboratory verification and has been used in many experiments. There are several papers we published using 18s rRNA (Yufan Tang et al., 2015; Yazhong Jin et al., 2016; Xiaoou Guo et al., 2017; Chong Zhang et al., 2017; Qiaojuan Xing et al., 2019).

3. Family genes have high sequence similarities and share several conserved motifs among them. As a result, specificity of primers is vital for the success of qRT-PCR analysis. The authors should give the detailed information on the method of primer design and provide the related supporting figures.

Re: We have followed your suggestion. The method of primer design is according to the reference below (Jian Yao et al., 2015). Then we uploaded the primers to NCBI website (https://www.ncbi.nlm.nih.gov/tools/primer-blast/) to check specificity of primers. Please see the related supporting information in S5 figure.

4. In the section of statistical analysis, the method used is not correct.

 Re: We are sorry for our incorrect expression. In the section of statistical analysis, significant differences were compared using a one-way ANOVA following Duncan’s multiple range test at p<0.05 level except expression profiles of CmMAPKs under RL induction were conducted by t test at p<0.05 level. We have revised it. Please see in the line 157-160 of the manuscript revision.

References:

1.Tang TF, Zhang C, Cao SX, Wang X, Qi HY. The Effect of CmLOXs on the Production of Volatile Organic Compounds in Four Aroma Types of Melon (Cucumis melo). PLOS ONE. 2015; 10(11). doi:10.1371/journal.pone.0143567

2.Jin YZ, Zhang C, Liu W, Tang YF, Qi HY, Chen H, et al. The Alcohol Dehydrogenase Gene Family in Melon (Cucumis melo L.): Bioinformatic Analysis and Expression Patterns. Frontiers in Plant Science.2016;7. doi:10.3389/fpls.2016.00670

3.Guo XO, Xu JJ, Cui XH, Chen H, Qi HY. iTRAQ-based Protein Profiling and Fruit Quality Changes at Different Development Stages of Oriental Melon.BMC Plant Biology.2017;17. doi:10.1186/s12870-017-0977-7

4.Zhang C, Cao SX, Jin YZ, Ju LJ, Chen Q, Xing QJ, et al. Melon13-lipoxygenase CmLOX18 may involved in C6 volatiles biosynthesis in fruit. Scientific Reports.2017;7. doi:10.1038/s41598-017-02559-6

5.Xing QJ, Zhang XL, Li YP, Shao QX, Cao SX, Wang F, et al. The lipoxygenase CmLOX13 from oriental melon enhanced severe drought tolerance via regulating ABA accumulation and stomatal closure in Arabidopsis. Environmental and Experimental Botany.2019; 103815.doi:10.1016/j.envexpbot.2019.103815

6.Yao J, Dong ZH, Ma XX, Wang XF. A simple method of real-time PCR primer design based on free online software. Genomics and Applied Biology. 2015; 34(12):2779-2784. doi:10.13417/j.gab.034.002779

---

## [Editor Report · Decision Letter 1]

22 Apr 2020

Genome-wide identification of mitogen-activated protein kinase (MAPK) cascade and expression profiling of CmMAPKs in melon (Cucumis melo L.)

PONE-D-20-01288R1

Dear Dr. Qi,

We are pleased to inform you that your manuscript has been judged scientifically suitable for publication and will be formally accepted for publication once it complies with all outstanding technical requirements.

With kind regards,

Yuan Huang

Academic Editor

PLOS ONE
---

## [Editor Report · Acceptance letter]

27 Apr 2020

PONE-D-20-01288R1 

Genome-wide identification of mitogen-activated protein kinase (MAPK) cascade and expression profiling of CmMAPKs in melon (Cucumis melo L.) 

Dear Dr. Qi:

I am pleased to inform you that your manuscript has been deemed suitable for publication in PLOS ONE. Congratulations! Your manuscript is now with our production department. 

With kind regards,

on behalf of

Dr. Yuan Huang 

Academic Editor

PLOS ONE